# Betulinic Acid Increases the Lifespan of *Drosophila melanogaster* via Sir2 and FoxO Activation

**DOI:** 10.3390/nu16030441

**Published:** 2024-02-01

**Authors:** Hye-Yeon Lee, Kyung-Jin Min

**Affiliations:** Department of Biological Sciences and Bioengineering, Inha University, Incheon 22212, Republic of Korea; lee.hy@inha.ac.kr

**Keywords:** betulinic acid, lifespan, physiology, anti-aging effect, *Drosophila melanogaster*

## Abstract

Betulinic acid (BetA), a triterpenoid derivative found abundantly in the plant kingdom, has emerged as a promising candidate for promoting longevity. Many research studies have shown its antioxidant, anti-inflammatory, antiviral, and anticancer activities, making it an interesting subject for investigating its potential influence on lifespan. This study aimed to investigate the effects of BetA on longevity and the mechanisms associated with it using the fruit fly *Drosophila melanogaster* as the organism model. The results showed that 50 μM BetA supplementation extended the mean lifespan of fruit flies by 13% in males and 6% in females without any adverse effects on their physiology, such as fecundity, feeding rate, or locomotion ability reduction. However, 50 μM BetA supplementation failed to increase the lifespan in mutants lacking functional silent information regulator 2 (Sir2) and Forkhead box O (FoxO)-null, implying that the longevity effect of BetA is related to Sir2 and FoxO activation. Our study contributes to the knowledge in the field of anti-aging research and inspires further investigations into natural compounds such as BetA to enhance organismal healthspan.

## 1. Introduction

In the quest for an extended and healthier lifespan, mankind has constantly attempted to find novel approaches. These range from advancements in medical science to simple lifestyle modifications. The efforts to extend lifespan and enhance healthspan (the period of life spent in good health and free from diseases) have focused attention on the challenges posed by an aging global population. Age-related diseases, such as metabolic syndrome, cardiovascular disorders, and neurodegenerative conditions, among many others, have become major public health concerns, increasing the need for novel therapeutic strategies to mitigate their impact. Consequently, identifying natural compounds that can alleviate age-related physiological decline and potentially extend lifespan has become a subject of growing interest. Among these compounds, betulinic acid (BetA, 3β-Hydroxy-20(29)-lupaene-28-oic acid, C_30_H_48_O_3_) [1], a triterpenoid derivative found abundantly in the plant kingdom, has emerged as a promising candidate for promoting longevity. BetA has a long history of traditional medicinal use, particularly in folk medicine, where extracts from various plants containing this compound have been employed to treat different diseases [2,3]. Bet A has been isolated and characterized from various plant sources, and a detailed analysis of its diverse pharmacological properties and molecular and cellular effects has been elucidated. BetA alleviates endoplasmic reticulum stress, which is crucial for cellular homeostasis and the prevention of diseases like nonalcoholic fatty liver disease [4]. It also protects against oxidative stress by enhancing the activation of the Nrf2/HO-1 pathway, which plays a critical role in cellular antioxidant responses, and inhibits the activation of stress pathways like p38 and JNK [5]. Moreover, BetA leads to reduced inflammation and modulation of the immune response by NF-κB inhibition [6] and induces programmed cell death by activating proapoptotic mitogen-activated protein kinase in human melanoma cells [7]. Extensive research in recent years has shown that it has antioxidant, anti-inflammatory, antiviral, and anticancer properties, making it an interesting subject for an investigation into its potential influence on lifespan [2,8,9]. Specifically, a recent study proved that BetA has an effect on longevity with an improvement in the healthspan due to its increased antioxidant activity and through its effects on the insulin/insulin-like growth factor-1 (IGF-1) signaling (IIS) pathway in *Caenorhabditis elegans,* a free-living nematode [10]. However, there is no evidence on whether BetA affects longevity in other organism models. This study aimed to investigate the effects of BetA on longevity and resistance to environmental stress in the fruit fly *Drosophila melanogaster*. Our study contributes to the knowledge in the field of anti-aging research and inspires further investigations into natural compounds such as BetA to enhance organismal healthspan.

## 2. Materials and Methods

### 2.1. Fly Husbandry

The lifespan assay was conducted using *Drosophila melanogaster (D.melanogaster)* Oregon-R, Dahomey, Canton-S, or white-eyed Canton-S (*w^CS10^*), a wild-type strain, which was purchased from the Bloomington *Drosophila* Stock Center in 2008 (BDSC, Indiana University, Indianapolis, IN, USA). All the following experiments were conducted using *w^CS10^*. The experiments were also carried out using mutant *D.melanogaster* strains, including *w^1118^*, *yw*, *w^1118^*;;*Sir2^2A-7-11^*, and tubulin-GAL4, which were obtained from the BDSC in 2008. The dominant negative form of the target of rapamycin (UAS-*TOR^FRB^*) and a dominant-negative S6k (UAS-*S6k^KQ^*) was provided by WJ Lee (SNU, Korea) in 2014. The *yw*;;*foxo^25^*/*TM6B Tb* and *yw*;;*foxo^21^*/*TM6B* were provided by K Yu (KRIBB, Korea) in 2018. The flies were cultured and reared at 25 °C and 65% humidity on 12:12 h light:dark cycles. Larval crowding was avoided by laying approximately 150 eggs on 250 cm^3^ fly bottles containing 25–30 mL of medium, which were developed until eclosion to the adult stage.

### 2.2. Food Preparation

A standard cornmeal–sugar–yeast (CSY) with agar medium was used to rear the fly larvae, and a standard sugar–yeast (SY) medium was provided with BetA when the flies were used in experiments. Betulinic acid (BetA, 3β-Hydroxy-20(29)-lupaene-28-oic acid, C_30_H_48_O_3_, Mw = 456.70) (Figure 1) [1] was purchased from Tokyo Chemical Industry (B2836, Tokyo, Japan). BetA was dissolved in 99% ethanol and added to the SY during food preparation at final concentrations of 10, 25, 50, or 100 μM. The control group (0 μM) was conducted by containing only the solvent (99% ethanol) in the medium. In the dietary restriction (DR) experiment, the concentration of brewer’s yeast in the SY media fed to separate groups of flies was changed to 2% yeast (DR diet) or 6% yeast (Full-fed diet). Appendix A lists the detailed recipes.

### 2.3. Lifespan Assays

Newly eclosed 100 male and 100 fertilized female adult fruit flies were transferred to a 500 cm^2^ demography cage using CO_2_ anesthesia. Three replicate cages were set up for each group (n = 300). Vials (Φ25 × 100 mm) containing fresh SY food with/without BetA were fixed to the cages and changed every two or three days. All dead flies were removed, and the numbers were recorded. The Kaplan–Meier survival estimator was used to estimate the survival function from lifetime data. The maximum lifespan is defined as the day by which the final 25% of a population remains alive. Log-rank tests were carried out to determine the statistical significance of the differences in the mean lifespan. The JMP statistical package (SAS Institute, Cary, NC, USA) was used for the analyses.

### 2.4. Physiological Experiments

Fecundity*:* Male and female fruit flies were collected separately every three hours after initiating the first eclosion. Female virginity was confirmed by the absence of progeny in the food after 24 h. The Φ25 × 100 mm vials were set up on day two, with a population density of two males and one virgin female. Every 24 h, the flies were transferred to new vials containing fresh SY food with/without 50 μM BetA. The number of eggs laid by each female daily was counted for 10 days. Twenty replicate vials were tested per treatment (total no. flies = 20). The statistical probability was determined using a *t*-test.

Feeding amount: The feeding rate measurement was conducted as described in [11]. After being fed the 50 μM BetA-supplemented diet for two weeks, 10 single-sex flies were transferred to a new Φ25 × 100 mm vial containing the same diet with the addition of a blue-dye number 1 (0.05% wt./vol). After feeding for 10 min, the anesthetized flies were washed with phosphate-buffered saline (PBS) and homogenized in 0.2 mL distilled water after head removal. After centrifugation for 5 min at 13,000 r/min (8400× *g*), the absorbance of the supernatants was measured at 595 nm using a spectrophotometer (Sunrise, TECAN, Männedorf, Switzerland). Three replicate vials were tested per treatment (total no. flies = 30). The statistical probability was determined using a Wilcoxon rank sum test.

Locomotion: To measure physical activity, the rapid iterative negative geotaxis protocol described by [12] is used for this test. Adult flies were fed diets containing 50 μM BetA for 0 to 4 weeks prior to performing the vertical climbing assay. Flies were collected under brief CO_2_ anesthesia (1–2 min) and allowed to recover for at least 18 h. Twelve single-sex flies were loaded into the apparatus for the vertical climbing assay and tapped on a tabletop three times in rapid succession to initiate negative geotaxis responses. The positions of the flies in the tubes were captured by digital images taken 4 s after initiating the behavior, and the number of flies up to 4 cm from the bottom was counted. In all the studies, the flies were assessed in consecutive trials separated by 1 min of rest, and four trials were used in all experiments. Fifteen replicate vials were tested per treatment (total no. flies = 300).

### 2.5. Body Weight

Newly eclosed adult flies were collected for two days. Ten flies were assigned under mild CO_2_ anesthesia and transferred randomly to a vial containing sterile SY food with 50 μM BetA supplement. Fifteen replicate vials were established for each group (total no. flies = 150). The vials were changed every three days for new vials containing fresh, sterile food. After 2 weeks, the body weight of the flies was measured on a microbalance (PAG214C, OHAUS, Parsippany–Troy Hills, NJ, USA) after CO_2_ anesthesia. The statistical probability was determined using a *t*-test.

### 2.6. Glutathione (GSH)/Glutathione Disulfide (GSSG) Ratio

Total glutathione (GSH) and glutathione disulfide (GSSG) were measured in a recycling assay using a commercially available GSH/GSSG kit (GT40, Oxford Biomedical Research, Rochester Hills, MI, USA). Briefly, twenty male flies were homogenized in an ice-cold 5% Metaphosphoric acid solution, and the supernatants were used for the measurement of total glutathione and GSSG by the evaluation of the extent of 2-nitro-5-thiobenzoic acid (DTNB) formation monitored at 415 nm for 10 min. The amounts of GSH and GSSG were calculated from a GSSG standard curve (μM), as detailed in the manufacturer’s instructions kit. Five replicates were established for each group (total no. flies = 100).

### 2.7. Triacylglyceride (TAG) Level

The level of TAG was measured as previously described [13]. Newly eclosed flies were pretreated with BetA 50 μM for 2 weeks. Heat-treated homogenized samples from 20 flies were used for the measurement of TAG and free glycerol using the TAG reagent (T2449, Sigma-Aldrich, St. Louis, MO, USA) and Free Glycerol Reagent (F6428, Sigma-Aldrich). At least three replicates were established for each group (total no. flies = 60), and the TAG level was normalized to the protein content.

### 2.8. Stress Resistance Test

The resistance to thermal stress, oxidative stress, and starvation of flies fed BetA were measured by improving the protocol described in previous studies [14,15,16].

Heat or cold shock: Thermal stress resistances were quantified as the time taken by adult flies to knock down or recover from exposure to heat and cold stress [14,15]. Flies were pretreated with 50 μM BetA for 2 weeks, and then ten single-sex flies were transferred into a new vial for the heat or cold shock test. For the heat shock resistance assay, flies were exposed to heat shock (38.5 °C) by dipping the vial into a water bath, and the number of deaths was recorded every 10 min until all the flies died. For the cold stress test, the flies were exposed to cold (0 °C) for 8 h and returned to room temperature, and then the minutes of recovery time until their awakening was measured. Fifteen replicates were established (n = 150). The statistical probability of average recovery time to cold shock was determined using a Wilcoxon rank sum test.

Oxidative stress: Ten 50 μM BetA-pretreated single-sex flies were fed medium supplemented with 5% sucrose and 18 mM paraquat (856177, methyl viologen dichloride hydrate, Sigma-Aldrich, St. Louis, MO, USA) [16]. Dead flies were scored every 6 h. Fifteen replicates were established (n = 150).

Starvation stress: Ten 50 μM BetA-pretreated single-sex flies were kept in vials containing 0.8% agar [16]. Dead flies were scored every 6 h. Fifteen replicates were established (n = 150).

The Kaplan–Meier survival estimator was used to estimate the survival function from lifetime data. Log-rank tests were carried out to determine the statistical significance of the differences in the mean lifespan. The JMP statistical package (SAS Institute, Cary, NC, USA) was used for the analyses.

### 2.9. Real-Time Quantitative PCR

After feeding the fruit flies with BetA for 5 weeks, the total RNA was extracted from the bodies of 15 male flies using RNAiso (Takara Bio, Kusatsu-Shi, Japan). The total RNA was reverse transcribed using M-MLV reverse transcriptase (Promega, Madison, WI, USA). Quantitative PCR was performed using the QuantStudio 3 (Applied Biosystem, Foster City, CA, USA) and TOPreal^TM^ qPCR 2× PreMix (Enzynomics, Daejeon, South Korea) according to the manufacturer’s instructions. At least three replicates were established for each group (n > 45), and all experiments were repeated at least three times. *Ribosomal protein 49* (*rp49*) was used as the internal control. Appendix A lists the primer oligonucleotide sequences. The statistical probability was determined using a *t*-test or the Wilcoxon rank sum test.

### 2.10. Statistical Analysis

Log-rank tests were carried out to determine the statistical significance of the survival analysis results. The JMP statistical package (SAS, Cary, NC, USA) was used for the analyses. The test for normality (Shapiro-Wilk test) and the statistical probabilities (F-test, *t*-test, or Wilcoxon rank sum test) of data in this study were performed using the R studio (v. 2022.12.0 + 353) software.

## 3. Results

### 3.1. Betulinic Acid Increases the Lifespan of Fruit Flies

To investigate the effects of BetA on the lifespan of fruit flies, the lifespans of four wild-type strains (Oregon-R, Dahomey, Canton-S, and white-eye Canton-S (*w^CS10^*)) were measured after treatment with BetA over a concentration range of 10 to 100 μM (Appendix A). BetA displayed an effect on the longevity of these four strains used in this study, indicating that BetA treatment may increase the longevity effect in most fruit fly strains. The optimum effect of BetA on longevity was observed in the *w^CS10^* strain (Appendix A). We repeated the survival test of the BetA-fed *w^CS10^* strain (Figure 2a,b, and Appendix A). The mean lifespan of the BetA-treated flies was increased compared to that of the non-treated flies (male, 0 μM, 44.84 ± 0.63 days; female, 0 μM, 42.31 ± 0.68 days). In males, the mean lifespans of the flies treated with BetA across all the BetA doses were significantly increased compared to the control group (10 μM, 49.96 ± 0.65 days, 11% increase, χ^2^ = 34.20, *p* < 0.0001; 25 μM, 47.84 ± 0.71 days, 7% increase, χ^2^ = 15.69, *p* < 0.0001; 50 μM, 50.82 ± 0.68 days, 13% increase, χ^2^ = 51.02, *p* < 0.0001; 100 μM, 47.73 ± 0.70 days, 6% increase, χ^2^ = 15.32, *p* < 0.0001). In females, the mean lifespan of flies treated with BetA at all doses except 100 μM BetA was significantly increased compared to control group (10 μM, 44.14 ± 0.75 days, 4% increase, χ^2^ = 6.63, *p* = 0.01; 25 μM, 44.11 ± 0.77 days, 4% increase, χ^2^ = 7.70, *p* < 0.01; 50 μM, 44.73 ± 0.79 days, 6% increase, χ^2^ = 12.07, *p* < 0.001; 100 μM, 42.58 ± 0.72 days, 1% increase, χ^2^ = 0.66, *p* = 0.4177). The highest extension of lifespan was observed at 50 μM BetA in both males and females. Moreover, age-related mortality was markedly lower in the male and female flies of all ages fed 50 μM BetA (Figure 2c,d, Appendix A). These results suggest that the BetA treatment extends the lifespan of fruit flies, and 50 μM BetA is the most effective concentration in the *w^CS10^* fruit fly. Accordingly, for a more detailed analysis of the effect of BetA, the combination of the *w^CS10^* strain and 50 μM of BetA was used in subsequent experiments.

### 3.2. Effects of BetA on Longevity Did Not Adversely Affect the Physiology of the Flies

As organisms age, physiological performance, such as reproduction, food intake, mobility, and development, suffers a functional decline. The fecundity, feeding rate, locomotor activity, and developmental viability of fruit flies fed BetA were measured to determine if the BetA treatment improved their healthspan. The BetA treatment tended to reduce the number of eggs laid by each female fly per day (Figure 3a), but the average number of eggs over 10 days was similar to that of the non-supplemented groups (Figure 3b, 0 μM, 27.07 ± 4.01; 50 μM, 23.96 ± 3.75, *t*-test, *p* = 0.11). The effects of the BetA treatment after egg hatching on the developmental viability of the larvae were investigated. It was found that the administration of BetA increased the ratio of pupation (Figure 3c, Larva-to-pupa, 0 μM, 88%; 50 μM, 95%, Wilcoxon rank sum test, *p* < 0.05) but did not affect the ratio of eclosion (Figure 3c, Pupa-to-adult, 0 μM, 72%; 50 μM, 65%, Wilcoxon rank sum test, *p* = 0.85). In the feeding behavior test, treatment with BetA resulted in reduced food intake in male flies (Figure 3d, Wilcoxon rank sum test, *p* < 0.05) but significantly increased the food intake in female flies (Figure 3d, Wilcoxon rank sum test, *p* < 0.05). In the locomotor activity test, the BetA treatment did not change the locomotor activity except for the two-week-old flies of both sexes, which showed increased activity (Figure 3e, left, male, *t*-test, *p* < 0.05; right, female, *t*-test, *p* < 0.0001). Interestingly, the body weight of the flies decreased when they were fed the food supplemented with BetA (Figure 3f, males, 8.1% decrease, *t*-test, *p* < 0.0005; females, 8.2% decrease, *t*-test, *p* < 0.0001), although the feeding rate was increased (Figure 3d). However, the lipid contents were decreased only in BetA-treated male flies but not in the females (Figure 3g, male, 41.0% decrease, *t*-test, *p* < 0.005; female, 5.4% increase, *t*-test, *p* = 0.59). Overall, these results suggest that the effects of BetA on longevity were not accompanied by any adverse effects on the physiological activity in fruit flies.

### 3.3. Treatment with BetA Did Not Change the Resistance to Thermal Stress and Oxidative Stress but Reduced the Resistance to Starvation Stress

Many pharmacological interventions used for their anti-aging effects increase the resistance to environmental stresses [17,18]. A recent study showed that supplementation with BetA increased the resistance to various environmental stresses in *C. elegans* [10]. To confirm that the beneficial effects of BetA against various stressors were conserved in the current organism model, the effects of BetA treatment on stresses, including heat shock, cold shock, oxidative stress, and starvation stress, were investigated (Figure 4). In our study, the BetA-treated flies did not show increased resistance to heat shock (Figure 4a, male, log-rank test, *p* = 0.4370; female, log-rank test, *p* = 0.5567). Under cold shock stress, the recovery time was unaffected in males (Figure 4b left, *t*-test, *p* = 0.1329) but was delayed in females after the BetA treatment (Figure 4b right, *t*-test, *p* < 0.001). Similar to the results of previous studies, the BetA treatment enhanced the resistance to oxidative stress in fruit flies (Figure 4c, males, log-rank test, *p* < 0.05; females, log-rank test, *p* < 0.05). The ratio of reduced glutathione (GSH) to oxidized glutathione (GSSG) within cells is a measure of cellular oxidative stress wherein a lower GSH-to-GSSG ratio is indicative of greater oxidative stress and the increase in GSH-to-GSSG ratio is indicative of resistance to oxidative stress. The BetA treatment resulted in an increase in the GSH-to-GSSG ratio and the gene expressions of antioxidant enzymes (Figure 4d,e). However, the BetA-treated fruit flies were more sensitive to starvation stress (Figure 4f, males, log-rank test, *p* < 0.05; females, log-rank test, *p* < 0.0001).

### 3.4. Longevity by BetA Supplementation Is Related to the DR-Mediated Longevity Effect

Even though there was no change in feeding rate both in males and females with BetA treatment (Figure 3d), the results showed that there was a reduction in body weight and fat accumulation (Figure 3f,g) and decreased starvation resistance (Figure 4f).

These features have also been observed in organisms with dietary restriction (DR). DR is the most effective intervention to induce longevity and provide a healthy lifestyle in all model organisms [19,20]. Several anti-aging agents have been reported to extend the lifespan of organisms via the regulation of mechanisms related to DR [21]. To determine whether the effects of BetA on longevity are related to DR, fruit flies were fed BetA in 2 or 6% of the brewer’s yeast diet media (Figure 5a,b and Appendix A and Appendix A). Interestingly, the BetA treatment still increased the lifespan of male flies even on a restricted diet (2% yeast diet) compared to the control group (Figure 5a, Appendix A, 6% yeast, vs. 0 μM; 12.5 μM, 12.14% increase, *p* < 0.0001; 50 μM, 12.14% increase, *p* < 0.0001; 2% yeast, vs. 0 μM; 12.5 μM, 4.88% increase, *p* < 0.05; 50 μM, 7.34% increase, *p* < 0.0001), but did not affect or decreased the lifespan of female flies under the DR diet (Figure 5b, Appendix A, 6% yeast, vs. 0 μM; 12.5 μM, 9.88% increase, *p* < 0.005; 50 μM, 6.27% increase, *p* < 0.05; 2% yeast, vs. 0 μM; 12.5 μM, 0.52% increase, *p* = 0.6183; 50 μM, 8.52% decrease, *p* < 0.05). These results suggest that the longevity effect of BetA is dependent on sex-specific differences and partially acts like DR.

The DR intervention induces the inhibition of the insulin/insulin-like growth factor signaling (IIS) pathway and the activation of the forkhead box O (FoxO) transcription factor [22]. To investigate whether the longevity effect of BetA is related to the IIS pathway and FoxO, this study evaluated the expression of insulin-like peptide genes and FoxO-target genes. Most expression levels of the insulin-like peptide genes except *dilp1*, *dilp2*, and *dilp6* and the FoxO-target genes did not change in the flies fed BetA for five weeks (Figure 5c, Wilcoxon rank sum test or *t*-test). However, the sirtuin (*sir2*) gene expression was significantly increased by BetA supplementation (Figure 5c, *t*-test, *p* < 0.005). Sirtuin has been shown to regulate longevity in numerous lower organisms, including fruit flies [23]. Thus, the longevity effect of BetA may be related to sirtuin activation.

To determine if the effect of BetA on longevity was mediated by sirtuin activation, the lifespans of flies completely lacking functional *sir2* (*w^1118^;sir2^4.5/5.26^*) and those of *dfoxo* null mutant (*yw;foxo^25/21^*) flies were measured with or without BetA treatment (Figure 5d–g and Appendix A). The results showed that wild-type *w^CS10^*, wild-type control flies (*w^1118^* and *yw*) fed BetA lived longer than the control flies not fed BetA (Figure 5d, *w^1118^* males, log-rank test, 6% increase, χ^2^ = 16.96, *p* < 0.0001; Figure 5e, *w^1118^* females, log-rank test, 14% increase, χ^2^ = 31.35, *p* < 0.0001; Figure 5f, *yw* males, log-rank test, 28% increase, χ^2^ = 47.84, *p* < 0.0001; Figure 5g, *yw* females, log-rank test, 7% increase, χ^2^ = 12.46, *p* < 0.0005). On the other hand, the lifespans of *sir2^4.5/5.26^* and *foxo^25/21^* flies were not increased by the BetA supplementation (Figure 5d, *sir2^4.5/5.26^* males, log-rank test, χ^2^ = 1.28, *p* = 0.2578; Figure 5e, *sir2^4.5/5.26^* females, log-rank test, χ^2^ = 0.17, *p* = 0.6791; Figure 5f, *foxo^25/21^* males, log-rank test, χ^2^ = 0.56, *p* = 0.4535; Figure 5g, *foxo^25/21^* females, log-rank test, χ^2^ = 0.00, *p* = 0.9513), suggesting that sirtuin and FoxO activation mediate the longevity effect of BetA. Taken together, the mechanism of the effect of BetA on longevity is similar to that seen due to DR, especially in terms of sirtuin and FoxO activation.

## 4. Discussion

Understanding the biological basis of aging and the potential effect of natural compounds such as BetA on lifespan holds promise not only for extending human lifespan but also for enhancing the overall quality of life during aging. Furthermore, exploring the mechanisms underlying the effects of BetA on longevity could offer valuable insights into the fundamental biology of aging and provide a basis for developing novel interventions to address the anti-aging effect. BetA is a natural compound found in the bark of several tree species, including the white birch tree, eucalyptus, plane trees, and mistletoe, as well as in certain fruits like berries [3,24,25]. BetA has been used as a traditional medicine for various diseases, and recently its effects on longevity were proven in *C*. *elegans*. However, there is still a lack of evidence on the effect of BetA on longevity in other organism models, and its mechanism remains unknown. In this study, we showed that treatment with BetA extended the lifespan of fruit flies without any adverse effects on physiological functions.

Moreover, the results of our study imply that the longevity effect of BetA is related to that of DR, especially with the activation of sirtuin and FoxO pathway. Interestingly, mistletoe (*Viscum album* L.), which is known to be rich in BetA, has also been shown to extend the lifespan in several animal models [26,27], and an earlier study proved that mistletoe extends the lifespan via FoxO activity induced by sir2 [28]. Therefore, other natural products containing BetA may also have a lifespan-extending effect.

Although the effect of BetA on longevity was observed in both sexes of fruit flies, some of the results on the physiology showed differences between males and females (Figure 3d,g, Figure 4b and Figure 5a,b). These results imply that the effect of BetA on physiology is partially sex-specific. For example, although the feeding behavior for females was enhanced with BetA treatment (Figure 3d), reduced fecundity in BetA-fed females (Figure 3a) might lower the body weight (Figure 3f) without reduction in the TAG levels (Figure 3g). Also, starvation resistance of flies fed BetA was reduced (Figure 4f), although no significant difference was observed in TAG levels in females. Our data showed that the feeding and locomotion abilities of fruit flies fed BetA tend to increase (Figure 3d,e), and previous studies revealed that BetA improved the metabolism [29,30]. It suggests the possibility that the increased metabolic activity by BetA may have increased their sensitivity to starvation stress. In Figure 5, the lifespan of males treated with BetA was still increased under DR conditions (Figure 5a white circles). On the other hand, the lifespan of females treated with BetA did not increase under DR conditions (Figure 5b white circles). Reduced fecundity can extend the lifespan of organisms and is also observed in organisms consuming a restricted diet [31,32]. These results suggest that BetA may extend the lifespan, perhaps through reduced fecundity in female flies. Interestingly, BetA failed to increase the lifespan of *sir2* or *foxo* mutant flies of both sexes (Figure 5d–g). Taken together, these results suggest that BetA supplementation may be used as a low-nutrient signal in fruit flies. Consistent with our results, a recent study using *C. elegans* showed that supplementation with BetA 50 μg/mL failed to increase the lifespan of *daf-16*, the *foxo* homolog in *C. elegans*, RNAi mutant worms [10]. Chen et al. (2022) reported that the longevity effect of BetA is related to the insulin signaling pathway in *C. elegans*. In this study, most of the insulin-like peptide gene expression was unchanged by BetA supplementation, but gene expression of *dilp1* and *dilp6* was increased, and gene expression of *dilp2* was decreased (Figure 5c). Interestingly, previous studies showed that increased expression of *dilp1* is also related to the reduction of reproductive diapause in females [33], and loss of *dilp2* can lead to longevity by up-regulation of *dilp1*, which promotes longevity as a pro-longevity insulin-like peptide [34,35]. Also, fat body-specific expression of dFoxO induces the downregulation of *dilp2* mediating by *dilp6* expression [36]. Adult fat body-specific expression of *dilp6* can extend lifespan [36], and dilp6 transcript levels are increased under malnutrition conditions [37]. Thus, these results also suggest that the longevity effect of BetA is related to DR and FoxO. Also, previous studies have shown the antioxidant and anti-obesity effects of BetA in rodent models [29,38,39,40,41]. Redox balance is closely connected to the upregulation of Sir2 and FoxO and the downregulation of IIS. Taken together, the longevity effect of BetA supplementation may be caused by mimicking the longevity effect of DR involving the regulation of Sir2, FoxO, and IIS pathways, which are known to regulate metabolism and redox balance (Figure 6).

## 5. Conclusions

In this study, we investigated the longevity effect and its mechanism of BetA as an anti-aging candidate. Our study reveals for the first time that the longevity effects of BetA are driven by the activation of Sir2 and FoxO transcription factors. Furthermore, when we consider all these studies, the beneficial effects of BetA on aging and healthspan have been observed in many organisms and could be applied to human healthspan. In conclusion, our investigation of BetA’s potential implications for extending the lifespan may give a direction for future research aimed at harnessing the therapeutic potential of natural compounds to promote healthy aging and longevity. This paper provides a direction for further investigations into natural compounds such as BetA to enhance healthspan in organisms.

## Figures and Tables

**Figure 1 nutrients-16-00441-f001:**
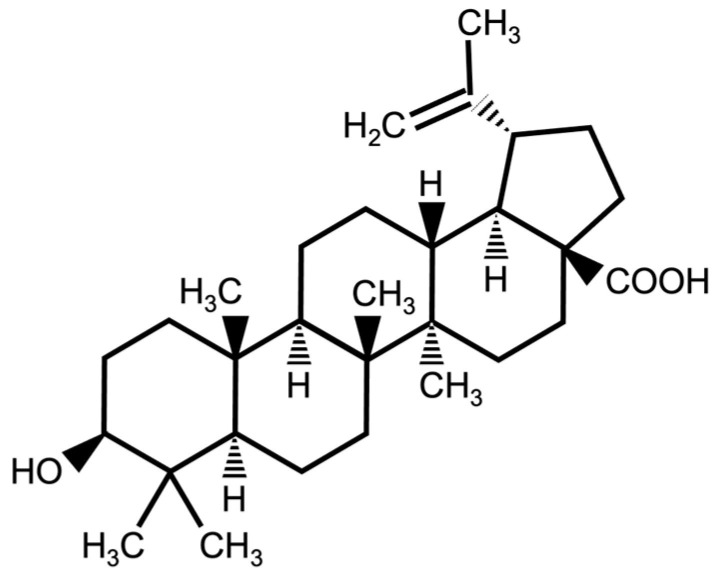
Chemical structure of betulinic acid (BetA).

**Figure 2 nutrients-16-00441-f002:**
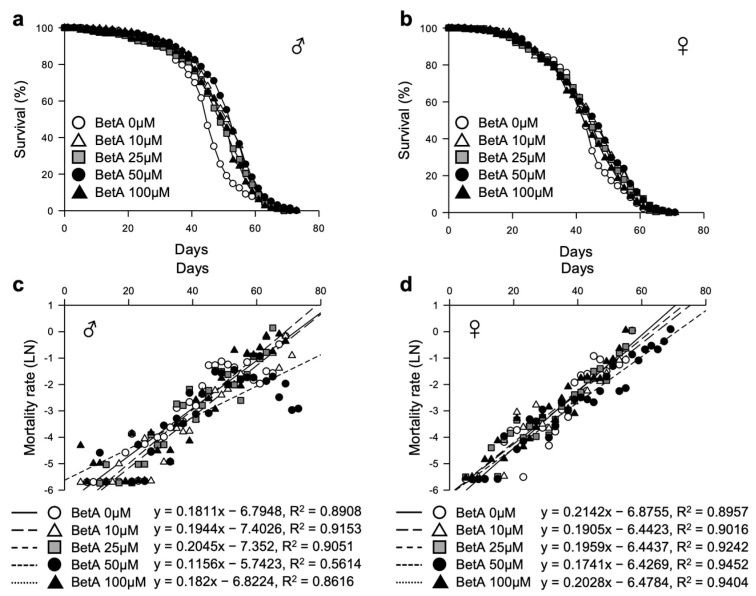
Effect of BetA on the lifespan of fruit flies. (**a**,**b**) Effect of BetA on the lifespan of males (**a**) and females (**b**). (**c**,**d**) Mortality curves of male (**c**) or female (**d**) flies supplemented with control or BetA. The natural log of the mortality rate was plotted using the Gompertz mortality model. BetA: Betulinic acid.

**Figure 3 nutrients-16-00441-f003:**
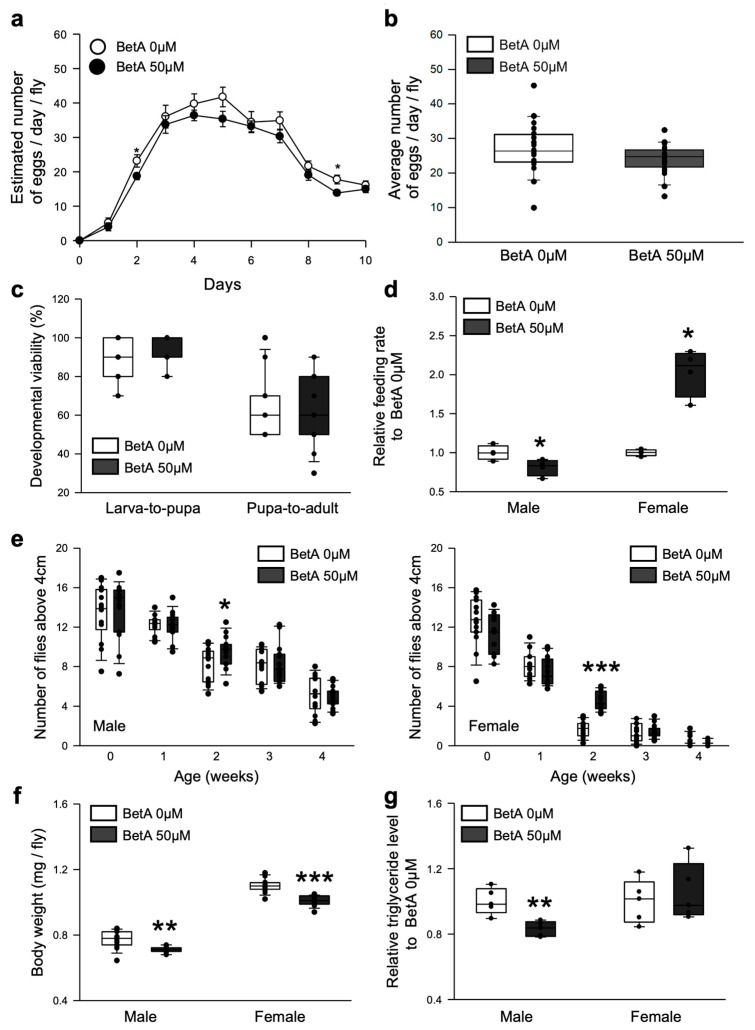
Effect of BetA on the physiology of fruit flies. (**a**) Number of eggs per day per female for 10 days (n = 20). (**b**) Average number of eggs laid by flies fed BetA for 10 days (n = 10). (**c**) The developmental viability of flies fed BetA or a control food (n = 15). Larval viability refers to the larva-to-pupa ratio, and pupal viability refers to the pupa-to-adult ratio. (**d**) The effects of BetA on the food intake of male (left boxes) and female (right boxes) flies for two weeks (n = 4). (**e**) The locomotor activity of male (left panel) or female (right panel) flies fed BetA or a control food. The locomotor activity of flies was measured every week for four weeks (n = 15). (**f**) Body weight of male (left boxes) or female (right boxes) flies fed BetA or control food for two weeks (n = 15). (**g**) Triacylglyceride (TAG) contents of male (left boxes) or female (right boxes) flies fed BetA or control food for two weeks (n = 5). The white symbols indicate the control group and the black symbols indicate the BetA-supplemented group. The statistical probability was determined using the *t*-test or the Wilcoxon rank sum test. The asterisks indicate significant differences compared to the control, * *p* < 0.05, ** *p* < 0.005, *** *p* < 0.0001. BetA: Betulinic acid.

**Figure 4 nutrients-16-00441-f004:**
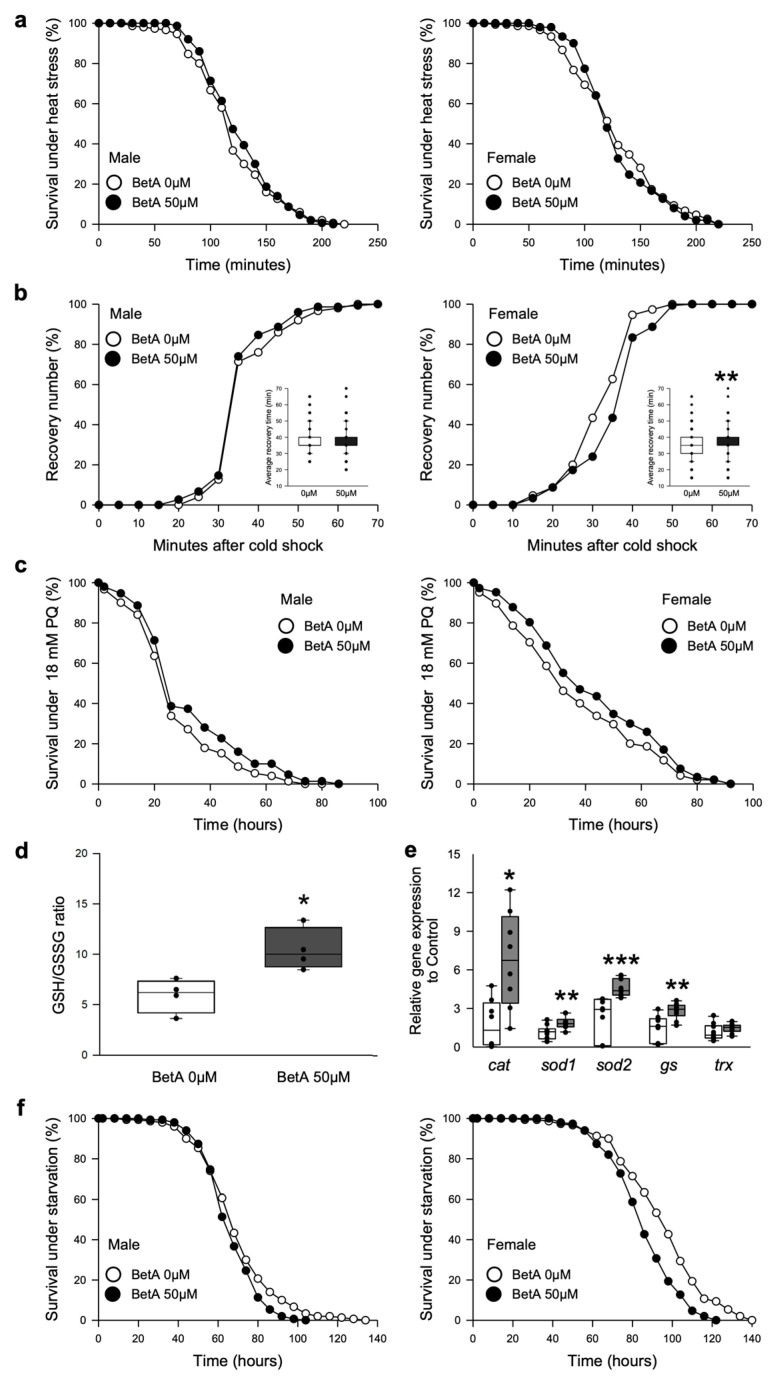
Effect of BetA on resistance to environmental stress of fruit flies. (**a**) The effects of BetA supplementation on resistance to heat shock of male (left panel) or female (right panel) flies fed BetA or a control food. (**b**) The effects of BetA supplementation on resistance to cold shock of male (left panel) or female (right panel) flies fed BetA or a control food. The embedded box plot indicates the average recovery time to cold shock. (**c**) The effects of BetA supplementation on resistance to 18 mM paraquat (PQ)-induced oxidative stress of male (left panel) or female (right panel) flies fed BetA or a control food. (**d**) The reduced glutathione (GSH)/oxidized glutathione (GSSG) ratio in the male fruit flies fed control or BetA food for two weeks (n = 4) (**e**) The mRNA levels of antioxidant enzyme genes were analyzed in the male fruit flies fed control or BetA food for two weeks (n = 9). Statistical probability was determined using the Wilcoxon rank sum test (*cat*, *sod2*, and *gs*) or *t*-test (*sod1* and *trx*). (**f**) The effects of BetA supplementation on resistance to starvation of male (left panel) or female (right panel) flies fed BetA or a control food. The white symbols indicate the control group and the black symbols indicate the BetA-supplemented group. The statistical probability was determined using the *t*-test or the Wilcoxon rank sum test. The asterisks indicate significant differences compared to the control, * *p* < 0.05, ** *p* < 0.005, *** *p* < 0.0001. BetA: Betulinic acid.

**Figure 5 nutrients-16-00441-f005:**
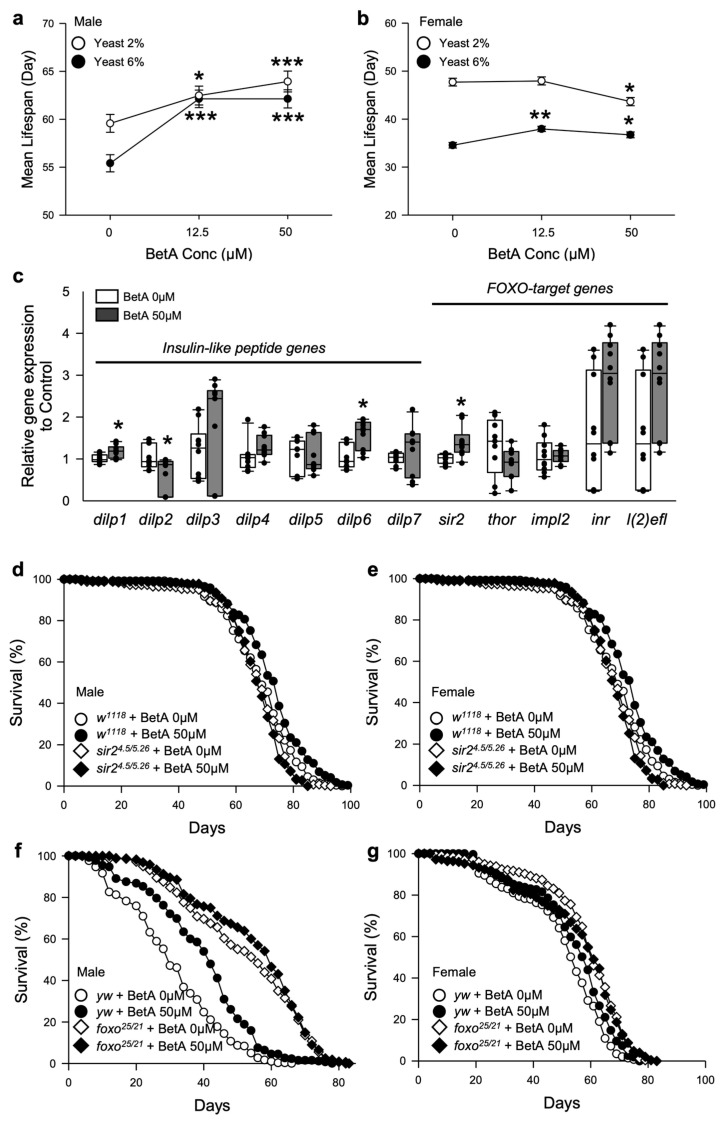
Relationship between the longevity effect of BetA and dietary restriction (DR) in fruit flies. (**a**,**b**) Mean lifespans of male flies (**a**) and female flies (**b**) fed BetA on 2, or 6% Brewer’s yeast diet. The white circles indicate the mean lifespan of flies fed a 2% yeast (DR) diet, and the black circles indicate the mean lifespan of flies fed a 6% yeast (Full-fed) diet. (**c**) The mRNA levels of insulin-like peptide genes and FoxO-target genes were analyzed in the male fruit flies fed a BetA-containing diet or a control diet for two weeks (n = 9). Statistical probability was determined using the Wilcoxon rank sum test (*dilp2-7*, and *l(2)efl*) or *t*-test (*dilp1*, *sir2*, *inr*, *impl2*, and *thor*). The white boxes indicate the lifespan of the flies fed food without BetA, and the black boxes indicate the lifespan of flies fed food with BetA. (**d**,**e**) Survival of *w^1118^* or *sir2^4.5/5.26^* mutant male flies (**d**) or female flies (**e**). (**f**,**g**) Survival of *yw* or *foxo^25/21^* mutant male flies (**f**) or female flies (**g**). The white symbols indicate the lifespan of the flies fed food without BetA, and the black symbols indicate the lifespan of flies fed food with BetA. The asterisks indicate significant differences compared to the BetA 0 µM. * *p* < 0.05, ** *p* < 0.005, *** *p* < 0.0001. BetA: Betulinic acid.

**Figure 6 nutrients-16-00441-f006:**
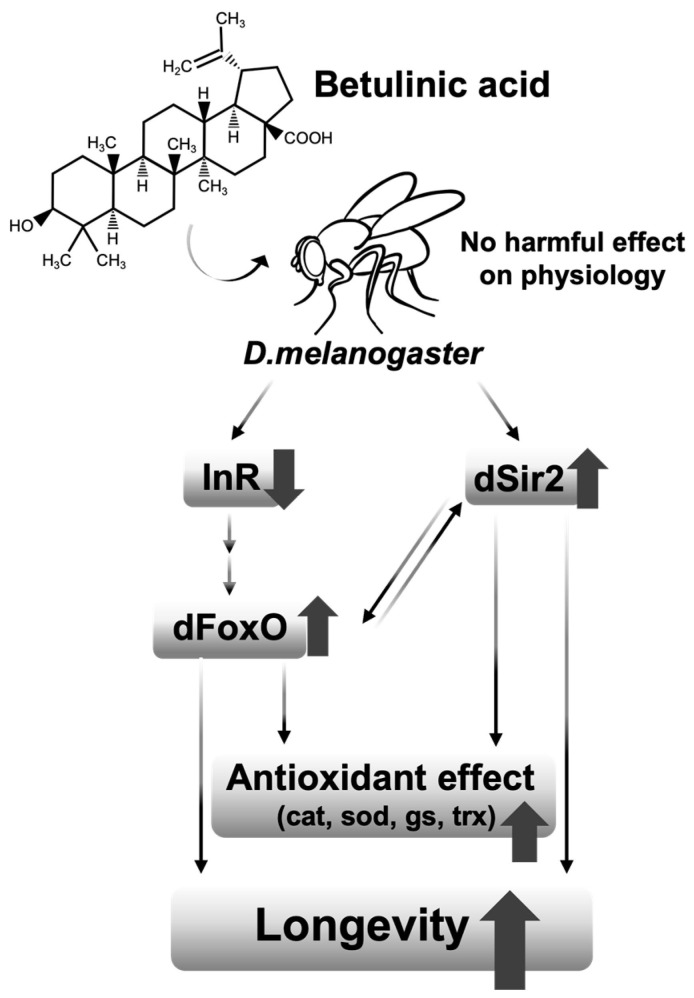
Comprehensive schematic diagram of longevity effect of BetA interacting with Sir2 and FoxO.

## Data Availability

Data are contained within the article and Appendix A.

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
