# Peer review of "Betulinic Acid Increases the Lifespan of *Drosophila melanogaster* via Sir2 and FoxO Activation"

_nutrients, 2024, doi:10.3390/nu16030441_

Round 1

Reviewer 1 Report

Comments and Suggestions for Authors

Lee & Min investigated here the effects of betulinic acid (BA) on longevity and protection against thermal and oxidative stress in fruit flies Drosophila.

The MS was well-written, clear on its objectives and hypothesis, and also presented a wide range of methods and techniques to accomplish its experimental goals. On the other hand, I still have some questions/suggestions before the acceptance of this paper:

(i) Please, include numeric data and overall conclusions within the Abstract. The abstract is too vague as it stands now.

(ii) Somewhere, preferably in an additional paragraph within the Introduction, present the chemical structure of BA and molecular/physiological mechanisms (or those already identified) by which BA could positively activate cellular protective responses (cytoprotective, antioxidant, etc)

(iii) Please, substantially improve method descriptions! There are many experimental protocols that were presented without a single reference!!!

For example: Feeding amount (lines 96-103), Locomotion (lines 104-112), Oxidative stress (lines 131-134), etc. Are those validated experimental methods? Method description is a MAJOR flaw in this paper.

(iv) Why some experimental assays were performed with merely N = 10 animals, sometimes N = 15, since we expect that sample size would not be an experimental obstacle when working with Drosophila flies? Accordinglt the authors presented experiments using N = 150, as we would expect.

(v) Figure 3: One of the most 'sound' results presented were the GSH/GSSG and mRNA expression of antioxidant enzymes sod, cat, gpx etc, which purportedly reinforces the free radical theory of aging, longevity etc. However, these important/key assays were performed with minimal N=3, N=4 samples. Again, the authors should think about a more robust bulk of data for such a nice work. Is it possible to increase the sample size in all aforementioned assays?

(vi) Please, include in the Discussion session a scheme showing the possible  interactions of BA with the FoxO pathway (or others) and reactive oxygen/nitrogen species or redox balance in cells.      

Comments on the Quality of English Language

I did not identify any typos or grammar errors, but I also recommend a double-check English revision

Author Response

Reviewer #1

Comment 1: Please, include numeric data and overall conclusions within the Abstract. The abstract is too vague as it stands now.

Response: We thank your comments on this point. We improved the abstract with several detailed numeric information.

Comment 2: Somewhere, preferably in an additional paragraph within the Introduction, present the chemical structure of BA and molecular/physiological mechanisms (or those already identified) by which BA could positively activate cellular protective responses (cytoprotective, antioxidant, etc).

Response: We thank your comments on this point. We added the reference showing the chemical structure of BetA in (page 1 Line 33 and page 2 Line 77) and some sentences presenting the molecular mechanisms of BetA on cellular protective responses (page 1 Line 40-47). 

Comment 3: Please, substantially improve method descriptions! There are many experimental protocols that were presented without a single reference!!!

For example: Feeding amount (lines 96-103), Locomotion (lines 104-112), Oxidative stress (lines 131-134), etc. Are those validated experimental methods? Method description is a MAJOR flaw in this paper.

Response: We thank your comments on this point. We added the relevant references about our experimental methods.

Comment 4: Why some experimental assays were performed with merely N = 10 animals, sometimes N = 15, since we expect that sample size would not be an experimental obstacle when working with Drosophila flies? Accordingly the authors presented experiments using N = 150, as we would expect.

Response: Because fruit fly mass is relatively small, experiments are conducted with many number of fruit flies in one sample. For instance, in the case of an experiment with 15 replications, one sample contains a minimum of 10 to a maximum of 20 flies, and the final number of fruit flies used is 150 to 300. To avoid confusion, we added the total number of flies used in the M&M section.

Comment 5: Figure 3: One of the most 'sound' results presented were the GSH/GSSG and mRNA expression of antioxidant enzymes sod, cat, gpx etc, which purportedly reinforces the free radical theory of aging, longevity etc. However, these important/key assays were performed with minimal N=3, N=4 samples. Again, the authors should think about a more robust bulk of data for such a nice work. Is it possible to increase the sample size in all aforementioned assays?

Response: We thank your kind comment on this point. We totally agree with you, but for these tests, we used 15-20 flies in one sample. Thus, the final number of fruit flies used in these tests is about 100. We added the total number of flies used in the M&M section. Additionally, considering that previous studies often set the number of samples to three (Aldrich & Maggert, 2014; Lee et al., 2015; Li et al., 2024; Matta et al., 2011; Smith et al., 2019), our study can be considered to have a relatively larger sample size than previous studies.

Comment 6: Please, include in the Discussion session a scheme showing the possible interactions of BA with the FoxO pathway (or others) and reactive oxygen/nitrogen species or redox balance in cells.

Response: We thank your critical comment on this point. We strengthened our discussion section with this point for your comment (page 11 Line 407-411).

Reviewer 2 Report

Comments and Suggestions for Authors

General comments

In this study, the effect of the plant-derived natural compound betulinic acid on the longevity of fruit flies was evaluated in addition to its effect on antioxidant, IIS signaling and sirtuin activity. The effect on feeding, reproduction, body mass and fat mass was also evaluated. Overall, they found that the median longevity of multiple strains was mildly, but significantly extended, in both sexes. There were no adverse effects on any of the physiological measures and antioxidant activity was generally increased. The beneficial effects of betulinic acid appears to require sirtuin and FOXO activity similar to what has been observed in dietary restriction models of longevity expansion in flies. However, although the median longevity was increased, there was no appreciable affect on the maximum longevity in this model which suggests the rate of aging itself was not affected.  Rather, the increase in longevity is likely attributable to beneficial effects on physiological function such as that seen with exercise.

Specific comments

1.      How was maximum lifespan calculated? Was it simply the age of the longest-lived individual?

2.      Although triglyceride levels were measured, there is no description of the method used.

3.      Why was the Canton-S strain chosen for further study when the effect seemed greatest in Oregon-R flies?

4.      On page 5, line 211 it is stated that the feeding rate was not changed although a significant difference is indicated on Figure 2. Moreover, the effect was different dependent on sex. Is this typical? Do female flies typically consume more than males normally?

5.      On Figure 2a, there are significant differences indicated for some days that seem to be the result of induvial t-tests for each day. This is not valid as each data point is not independent and there was no correction made for multiple comparisons. Why weren’t these data analyzed using an repeated measures ANOVA or a similar model?

6.        Could the observed difference in starvation resistance have to do with differences in lipid reserves among the treatment groups, especially females?

7.      Why were mean longevities used when comparing the effect of Beta on flies subjected to DR rather than log-rank tests?

8.      In addition to Sirt2, there was a significant effect of Beta on several dilps. Why were these effects ignored?

9.      Why weren’t the sirt2 and foxo mutants also subjected to DR in parallel as controls in this study?

10.   Figure 4f and 4g. Is the dramatic difference in survival curves between male and female foxo mutants typical? Has this been reported in other studies?

11.   The effect of Beta in virgin and/or sterile flies is needed before claims about differential fecundity can be made. 

Author Response

Reviewer #2

Comment 1: How was maximum lifespan calculated? Was it simply the age of the longest-lived individual?

Response: The Kaplan–Meier survival estimator was used to estimate the maximum lifespan. The JMP statistical package was used for the analyses. The maximum lifespan in this study is when the last 25% of the fruit flies used in the experiment are alive. Thus, in this study, maximum lifespan was referred to as the greatest age reached by 25% member of a given population. We wrote this information in M&M section (page 2 Line 90-91).

Comment 2: Although triglyceride levels were measured, there is no description of the method used.

Response: We thank your comments on this point. We added this description in M&M section (page 3 Line 142).

Comment 3: Why was the Canton-S strain chosen for further study when the effect seemed greatest in Oregon-R flies?

Response: We presented representative experimental results in the figure S1. Three repeated lifespan test was conducted with Oregon-R (OR) or CW strain. As a result, in the case of OR, effect of increasing lifespan by BetA was not observed in 1 out of 3 trials, but in the case of CW, an increase in lifespan by BetA was confirmed in all 3 trials. Therefore, that’s why that we selected CW, which showed a stable lifespan extension effect by BetA, and conducted additional experiments.

Comment 4: On page 5, line 211 it is stated that the feeding rate was not changed although a significant difference is indicated on Figure 2. Moreover, the effect was different dependent on sex. Is this typical? Do female flies typically consume more than males normally?

Response: We thank your comments on this point. We modified the words from ‘did not change’ to ‘were increased’ (page 6 Line 240). Also, it is typical in fruit flies that females have greater food consumption than males (Wong et al., 2009).

Comment 5: On Figure 2a, there are significant differences indicated for some days that seem to be the result of induvial t-tests for each day. This is not valid as each data point is not independent and there was no correction made for multiple comparisons. Why weren’t these data analyzed using an repeated measures ANOVA or a similar model?

Response: We performed the test of normality in all our data with Shapiro–Wilk test and determined statistical hypothesis. In the case of fecundity test, normality was conducted at each day of age or at the overall average over 10 days, and all results were determined as normal distribution. Since the comparison is only done with the control group, we decided on a t-test.

Comment 6: Could the observed difference in starvation resistance have to do with differences in lipid reserves among the treatment groups, especially females?

Response: We thank your critical comments on this point. In general, high TAG level in the body may increase resistance to starvation stress. However, in our female results, starvation resistance of flies fed BetA was reduced although no significant difference was observed in TAG levels. Our data showed that the feeding and locomotion abilities of fruit flies fed BetA tend to increase (Figures 2d and 2e) and previous studies revealed that the BetA improved the metabolism (Kim et al., 2019; Song et al., 2021). It suggests possibility that the increased metabolic activity by BetA may have increased their sensitivity to starvation stress. We added some sentences about this issue in Discussion section (page 11 Line 380-385)

Comment 7: Why were mean longevities used when comparing the effect of Beta on flies subjected to DR rather than log-rank tests?

Response: We used the same analysis method to confirm the lifespan extension effect of BetA with DR. Statistical significance was confirmed through log-rank test. To make it visually easier for readers, only the mean lifespan is used to the figure. We added the DR survival graph as Figure S2.

Comment 8: In addition to Sirt2, there was a significant effect of BetA on several dilps. Why were these effects ignored?

Response: We thank your critical comment on this point. We strengthened our discussion section with this point for your comment (page 9 Line 320, page 11 Line 395-406).

Comment 9: Why weren’t the sirt2 and foxo mutants also subjected to DR in parallel as controls in this study?

Response: In these data, we aimed to find molecular targets related to the lifespan extension effect of BetA by experimenting with targets that are well known to be associated with longevity and DR/CR. Thus, because we conducted an experiment to determine the relationship between sir2 or FoxO and the lifespan extension effect of BetA, experiments conducted with DR were excluded.

Comment 10: Figure 4f and 4g. Is the dramatic difference in survival curves between male and female foxo mutants typical? Has this been reported in other studies?

Response: As you observe, differences of survival curve between female and male are typically showed in foxO mutants. This difference has been reported in previous study (Fink et al., 2016).

Comment 11: The effect of Beta in virgin and/or sterile flies is needed before claims about differential fecundity can be made.

Response: Unfortunately, in the case of fruit flies, unmated fruit flies have little ability to produce eggs (approximately 0-1 egg/day). Therefore, the effect of BetA on fecundity cannot be compared in unmated fruit flies.

Reviewer 3 Report

Comments and Suggestions for Authors

The author claims in the abstract that:

 “However, there is still a lack of evidence on the effects of BetA on longevity” which is not true as there are many reports addressing that and author should address them.

Also, it is well known that “Sir2 and longevity: The p53 connection” and “FOXO Transcription Factors in the Regulatory Networks of Longevity

So, this paper lacks novelty and the author needs to highlight what is novel about this article.

There is no + ve control in most of the experiments.

Comments on the Quality of English Language

The author claims in the abstract that:

 “However, there is still a lack of evidence on the effects of BetA on longevity” which is not true as there are many reports addressing that and author should address them.

Also, it is well known that “Sir2 and longevity: The p53 connection” and “FOXO Transcription Factors in the Regulatory Networks of Longevity

So, this paper lacks novelty and the author needs to highlight what is novel about this article.

There is no + ve control in most of the experiments.

Author Response

Reviewer #3

Comment 1: “However, there is still a lack of evidence on the effects of BetA on longevity” which is not true as there are many reports addressing that and author should address them.

Response: We thank your comments on this point. We improved our abstract following your opinion.

Comment 2: It is well known that “Sir2 and longevity: The p53 connection” and “FOXO Transcription Factors in the Regulatory Networks of Longevity”

So, this paper lacks novelty and the author needs to highlight what is novel about this article.

Response: In this study, we investigated the lifespan extension effect and mechanism of BetA, an anti-aging candidate. Our study reveals for the first time that the lifespan-extending effects of BetA are driven by activation of sir2 and FOXO transcription factors. We added a sentence emphasizing the novelty of our study (page 11 Line 413-415).

Comment 3: There is no + ve control in most of the experiments.

Response: We expressed 0 μM BetA as a control in each experiment, which is equivalent to the +ve control because it contains only the solvent (99% ethanol). To avoid confusion, we added the explanation of control in the M&M section (page 2 Line 79-80).

Round 2

Reviewer 1 Report

Comments and Suggestions for Authors

The authors provided an improved verson of theur original MS but, still, there are MINOR issues that have to be fixed before its acceptance:

(i) please, include the chemical structure (the figure), not simply the chemical description of the molecule. The figure would help Nutrients' readers to understand how BA would molecularly interact with the cellular protective signal cascades.

(ii) I insist that a comprehensive figure of BA potentially interacting with FoxO and sir2 cascades (activation mechanism) in Discussion would trigger readers interest for revealing more molecular aspects of BA and its protective role in living organisms. That would increase the number of citations of this article and the journal, in general

Author Response

revised as suggested

Reviewer 2 Report

Comments and Suggestions for Authors

The authors have adequately addressed my concerns.  I have no further comments/questions. 

Author Response

Thank you